# Structural and Mechanical Properties of a-BCN Films Prepared by an Arc-Sputtering Hybrid Process

**DOI:** 10.3390/ma14040719

**Published:** 2021-02-03

**Authors:** Yuki Hirata, Ryotaro Takeuchi, Hiroyuki Taniguchi, Masao Kawagoe, Yoshinao Iwamoto, Mikito Yoshizato, Hiroki Akasaka, Naoto Ohtake

**Affiliations:** 1Institute of Innovative Research, Tokyo Institute of Technology, 4259 Nagatsuta, Midori-ku, Yokohama, Kanagawa 226-8503, Japan; ohtake.n.aa@m.titech.ac.jp; 2Department of Mechanical Engineering, Tokyo Institute of Technology, 2-12-1, O-okayama, Meguro-ku, Tokyo 152-8550, Japan; takein2ryo51310@ezweb.ne.jp (R.T.); hiroyuki229k@gmail.com (H.T.); mk94325@ezweb.ne.jp (M.K.); iwamoto.y.ag@m.titech.ac.jp (Y.I.); yoshizato.m.aa@m.titech.ac.jp (M.Y.); akasaka.h.ac@m.titech.ac.jp (H.A.)

**Keywords:** a-C film, a-BCN film, vacuum arc deposition, magnetron sputtering, structural properties, mechanical properties

## Abstract

Amorphous boron carbon nitride (a-BCN) films exhibit excellent properties such as high hardness and high wear resistance. However, the correlation between the film structure and its mechanical properties is not fully understood. In this study, a-BCN films were prepared by an arc-sputtering hybrid process under various coating conditions, and the correlations between the film’s structure and mechanical properties were clarified. Glow discharge optical emission spectroscopy, X-ray photoelectron spectroscopy, Fourier-transform infrared spectroscopy, and Raman spectroscopy were used to analyze the structural properties and chemical composition. Nanoindentation and ball-on-disc tests were performed to evaluate the hardness and to estimate the friction coefficient and wear volume, respectively. The results indicated that the mechanical properties strongly depend on the carbon content in the film; it decreases significantly when the carbon content is <90%. On the other hand, by controlling the contents of boron and nitrogen to a very small amount (up to 2.5 at.%), it is possible to synthesize a film that has nearly the same hardness and friction coefficient as those of an amorphous carbon (a-C) film and better wear resistance than the a-C film.

## 1. Introduction

Energy-saving and environment-friendly technologies have gained increasing attention in recent years from the viewpoint of preventing global warming and the depletion of energy resources. The friction loss in the sliding parts of automobiles and machine tools is estimated to be of the order of several hundreds of billion Euros per year [1]. Owing to the problem of environmental pollution, the use of lubricant oils and grease between sliding parts or during machine part cutting is being minimized. Thus, to reduce friction loss and contribute to energy saving, non-lubricating sliding and non-lubricating cutting technologies have been explored. Among these, hard coating technology has attracted significant attention because such functional thin films have excellent mechanical properties, such as high hardness, low coefficient of friction, and high wear resistance. In many cases, these hard films are composed of light elements like carbon, nitrogen, and boron, and they are widely applied to moving parts to reduce friction loss and extend the service life of machine tools [2,3]. Diamond-like carbon (DLC), cubic boron nitride (c-BN), and amorphous boron carbon nitride (a-BCN) are well known examples of hard coatings. DLC films are widely used in various applications because of their unique properties [4]. However, the heat resistance of DLC film depends on its structure; it has been reported that the heat resistance of hydrogenated amorphous carbon film (a-C:H) is ~300–350 °C [5,6], and that of tetrahedral amorphous carbon film (ta-C) is ~400–600 °C [7]. These values are considerably lower than that of c-BN film, ~1000–1360 °C [8,9]. In addition, DLC films cause diffusion wear against iron-based metals, and, hence, they cannot be coated on cutting tools for processing steel materials.

To overcome these shortcomings, the application of a-BCN films on iron-based material has been explored in recent years. a-BCN film has nearly the same properties as the DLC film, is also inert against steel material [10,11], and has an even superior tribological property [12,13] and thermal stability [14,15,16]. In particular, a weak dependence on the relative humidity of wear and friction coefficients [13], and excellent sliding characteristics even in a high temperature environment of 300 °C [16] are incredible features of a-BCN film as compared to DLC film. Certainly, it is true that c-BN has advantages in terms of heat resistance or inertness against steel material as compared to a-BCN film, which is relatively easy to synthesize, whereas high temperature and high pressure are required to synthesize c-BN. Chemical vapor deposition (CVD) and physical vapor deposition (PVD) are the mainly used techniques for the synthesis of a-BCN films. In the CVD process, a-BCN film is deposited on the substrate via a vapor-phase chemical reaction process. Many researchers have synthesized a-BCN film using many kinds of CVD processes, such as thermal CVD [17] and cold wall CVD [18] plasma CVD methods [19,20] under various coating conditions. However, because hydrogen-based gas is usually used as a raw material in the CVD process, these coatings are often soft (owing to the presence of hydrogen in the film structure). For example, according to Thamm et al., the hardness of a-BCN films with a high hydrogen content shows a value of around 13 to 15 GPa, and it increases as the hydrogen content decreases [19]. On the other hand, because the PVD method uses a solid material as raw material, it is possible to synthesize hydrogen-free hard films safely and easily without using toxic or flammable gases, as in the CVD method.

Recently, some studies related to physical vapor-deposited a-BCN films have been reported. Zhou et al. deposited a-BCN film by radio frequency magnetron sputtering from hexagonal boron nitride and graphite targets in an Ar–N_2_ gas mixture. They investigated the effect of the voltage applied to the substrate on the atomic composition ratio, as well as the hardness and the friction coefficient of the film [21]. Yasui et al. also studied the effect on the elemental composition ratio, film hardness, and friction coefficient of a-BCN film. They prepared the film by the ion-beam-assisted vapor deposition method while changing the sputtering time [22]. Apart from these researches, related to mechanical property evaluations, Todi et al. investigated the correlation between the optical properties and structural properties of the a-BCN film by applying the annealing treatment to the prepared sample [23]. Researching a new deposition method, Yoshida et al. aimed at forming a-BCN films using a sputtering-PBII hybrid system. They discussed the correlation among B–C–N compositions, the structure in the film, and the mechanical characteristics. The bonding state was evaluated by FT-IR, and as for the mechanical properties, a nanoindentation hardness test, an internal stress measurement, and a frictional wear test were conducted [24]. As these studies show, there is wide research on a-BCN films formed by the PVD method, but almost all studies focus individually on the film composition, the mechanical/optical properties of the film, or the film structure. In connection with this research, Yoshida and coworkers tried to clarify the correlation among the composition, properties, and structure of the film [24], but the types of a-BCN samples they prepared were very limited, and the research results cannot be said to be sufficient yet. To expand the application range of a-BCN films, it is necessary to correlate and systematize the structure and its mechanical properties. The objective of this study was to clarify and systematize the correlation between the structural and mechanical properties by evaluating the elemental composition ratio, bonding state, hardness, friction coefficient, and wear volume of the prepared a-BCN film.

## 2. Experimental

### 2.1. Deposition

The a-BCN film was prepared by a PVD method newly developed for this research. It combines vacuum arc deposition and magnetron sputtering (arc-sputtering hybrid process). The schematic of the arc-sputtering hybrid process is shown in Figure 1. The strength of the arc-sputtering hybrid process is the installation of an arc gun as a vacuum vapor deposition method. This enables the evaporated particles to have a high ionization rate and a high kinetic energy to bombard the substrate under an ultra-high vacuum environment. Furthermore, because the arc gun is arranged diagonally to the substrate on which the film is deposited, droplet deposition is reduced when compared to that in the conventional vacuum arc process. A pure graphite target or a graphite target containing boron (5 to 20%) was used as the target of the arc gun. Hexagonal boron nitride (h-BN) was used as the target of the sputtering gun. Table 1 lists the process parameters used for the deposition of the a-BCN film. Films with various compositions and structures were prepared by varying the gas flow ratio of nitrogen and argon, radio frequency power for magnetron sputtering, and power supply to the arc gun. Furthermore, to elucidate the effect of the incident energy of carbon ions on the film quality, the substrate potential was set as floating or −100 V. The bias voltage of −100 V was set because it is known that ions need to be incident at 100 eV to make the film with the highest sp^3^ binding ratio [4]. For comparison, a-C film was also synthesized by vacuum arc deposition. In this study, the measurement point of the film properties was set at the irradiation center position of the arc gun, as shown in Figure 1.

### 2.2. Film Evaluation

#### 2.2.1. Structural Properties

Transmission electron microscopy (TEM) studies were conducted to analyze the cross-section of the a-BCN film. The acceleration voltage of TEM was 200 kV. X-ray photoelectron spectroscopy (XPS) was used to analyze the chemical composition of the deposited films. For the XPS measurements, the films were etched with argon ions to remove surface contamination. A Gaussian–Lorentzian peak fitting of the spectra was performed after a Shirley background correction to evaluate the elemental composition of the film. In addition, glow discharge optical emission spectroscopy (GDOES) was used simultaneously to evaluate the elemental distribution in the depth direction. Fourier-transform infrared spectroscopy (FTIR) was used to evaluate the chemical bonding state. To reduce the possible influence of the moisture caused by carbon dioxide and water vapor in the atmosphere, FTIR measurements were conducted after performing a nitrogen purge to thoroughly dry the samples and obtain a more stable environment. The measurement range was 400–4000 cm^−1^, and the resolution was 4 cm^−1^. Furthermore, the structure of the film was analyzed using Raman spectroscopy. Normally, the Raman spectrum of carbon-based materials has a G peak near 1580 cm^−1^ corresponding to the six-membered ring of graphite and a D peak near 1360 cm^−1^ corresponding to the disorder at the end of the six-membered ring of the graphite structure [4,25]. By estimating these peak parameters, i.e., the intensity ratio of the D peak to the G peak (I(D)/I(G)) or the full width at half maximum of the G peak (FWHM(G)), and the shift of the G peak position, it is possible to evaluate the structure of the film. The wavelength of the laser was 532 nm, and the intensity was 0.34 mW. The laser exposure time was 1 s, and the number of accumulations was set to 20. The obtained spectrum was separated into G and D peaks using the Gaussian function.

#### 2.2.2. Mechanical Properties

The hardness of the a-BCN film was measured by nanoindentation. The indentation test was conducted under the following conditions: maximum load of 30–40 µN, loading time of 10 s, maximum load holding time of 1 s, and unloading time of 10 s. The number of measurement points was 49, and the hardness value was obtained by taking their average. In this test, the penetration depth was controlled using a vertical displacement meter to avoid substrate effect.

To evaluate the sliding characteristics, ball-on-disk tests were performed. Throughout the ball-on-disk test, the coefficient of friction and the specific wear of the ball and the disc sample were estimated simultaneously. The wear volume of the ball sample, *V*_ball_ (m^3^), was calculated using Equation (1), where *a* is the short axis, *b* is the long axis, and *D* is the diameter of the sphere.
(1)$$V_{ball}=\frac{\pi a^{3}b}{32D}$$

The wear volume of the disk sample *V_disk_* (m^3^) was calculated using Equation (2) from the wear area *S_disk_* (m^2^) and the sliding radius *R* (m). The cross-section of the wear mark was measured using a laser microscope.
(2)$$V_{disk}=2\pi RS_{disk}$$

By dividing the respective wear volumes *V* by the load *F* (N) and the sliding distance *L* (m), the specific wear rates of the ball and the disk *W* (m^3^/N·m) were derived, as shown in Equation (3).
(3)$$W_{ball}=\frac{V_{ball}}{FL},\;W_{disk}=\frac{V_{disk}}{FL}$$

A bearing steel ball (Japan standard, SUJ2) with a diameter of 6 mm was used as the ball sample. The sliding radius was 3 mm, the load was 1 N, and the sliding speed was 400 rpm. The test was carried out under dry sliding conditions. Since the coefficients of friction of a-C and a-BCN films, such as temperature and humidity [3,26,27,28], depend on the sliding environment, the sliding wear tests were conducted under the same environmental temperature (room temperature, 22–26 °C) and humidity of 22–25%.

## 3. Results and Discussion

### 3.1. Structural Properties

#### 3.1.1. TEM and GDOES Studies: Systematizing Based on the B–C–N Ternary Diagram

Figure 2 shows the cross-sectional TEM images of the a-BCN films obtained at low (Figure 2a) and high magnifications (Figure 2b). The thickness of the film is approximately 200 nm. The a-BCN film was found to adhere well to the silicon substrate. The lattice structure of silicon is observed, and there was no other lattice fringe. Therefore, no structural heterogeneity due to intermittent carbon irradiation of the arc gun is observed.

The chemical composition of the a-BCN films was measured by GDOES. Figure 3 shows the depth profiles of the a-BCN films. The carbon content was approximately 90%, and the elemental ratio of nitrogen and boron was <10%. The depth profile obtained by GDOES also indicated that the composition of the films was uniform throughout the thickness. Figure 4 shows the elemental composition of the a-BCN film prepared in this study and that of the a-C film prepared by the vacuum arc deposition method plotted on a B–C–N ternary diagram. It is evident that the a-BCN film was synthesized with a carbon content of approximately 45–100%. In addition, the a-BCN films synthesized by the arc-sputtering hybrid process can be categorized into three groups depending on the ratio of boron and nitrogen, viz. 4:1, 2:1, and 1:1, and it can be confirmed that the amount of carbon contained in the film is limited according to the ratio of boron and nitrogen. For instance, the sample with the ratio of boron and nitrogen is 4:1, and the a-BCN film could not be synthesized unless the carbon content was 60% or more. Then, as the ratio of boron and nitrogen became 2:1 and 1:1, it was possible to synthesize an a-BCN film with a smaller amount of carbon. The effects of carbon content on the structural and the mechanical properties are discussed in the following sections.

#### 3.1.2. FTIR Studies

Figure 5 shows the IR spectra of the a-BCN films. Broad peaks around 1000 to 1700 cm^−1^ are observed when the carbon content is <90%, whereas there is no noticeable peak when the carbon content is >90%. The broad peaks are a combination of the h-BN-bond-derived peak at 1380 cm^−1^, the CN-bond-derived peak at 1300 cm^−1^, and the BC-bond-derived peak at 1070–1250 cm^−1^. A weak absorption band near 1050 cm^−1^ is observed in the film containing more than 90% carbon. This indicates the presence of small amounts of cubic boron carbon nitride (c-BCN) phase in the film. With the increase in the nitrogen content, a peak corresponding to the C=N bond at 1600 cm^−1^ and a peak corresponding to the C≡N bond at 2200 cm^−1^ are observed. At a boron and nitrogen ratio of 4:1, the center position of the peak shifts from approximately 1300 to 1400 cm^−1^ as the carbon content decreases. In other words, B–C and C–N bonds are initially observed, but as the carbon content decreases, the amount of B–C and C–N bonds tends to decrease and that of hBN tends to increase. On the other hand, when the boron and nitrogen ratios were 2:1 and 1:1, there was no significant shift in the peak positions; all the peaks were centered near 1400 cm^−1^, which corresponds to the peak position of hBN. Furthermore, the peaks become sharper as the carbon content decreases, indicating that boron and nitrogen formed stable hBN bonds. This is because the atomic binding energies of B–C bonds (448 kJ) and C–N bonds (757 kJ/mol) are higher than that of B–N bonds (389 kJ/mol). Hence, B–N bonds are easily broken by C ions, and sharp peaks of hBN appear as the relative content of boron and nitrogen to carbon increases. Previous studies using FTIR analysis on BCN films also reported that the B–N bonding state is the most stable. They concluded that the hierarchy of energetic stability of bonds in BCN follows the sequence B–N > C–C > C–N > C–B > B–B > N–N. The same conclusion was reached despite the different depositions method used in these studies, namely plasma CVD [20,29], ion-beam-assisted deposition [22,30], magnetron sputtering [31], pulse laser deposition [32], or laser abration [33].

A Gaussian fitting of the peaks in the range of 1300–1500 cm^−1^ was carried out for samples with carbon content <5%. An example of the fitted spectrum is shown in Figure 6a. Figure 6b shows the results of the peak intensity ratio of hBN and B–O (I(hBN)/I(B–O)) and the full width at half maximum of the hBN peak (FWHM(hBN)) for each sample. It is evident that the closer the B:N ratio is to 1, the higher the I(hBN)/I(B–O) and the smaller the FWHM(hBN). This indicates that the crystallization of the hBN structure has actually progressed. This peak fit result shows that the FWHM (G) when the ratio of boron and nitrogen is 1: 1 was about 55% compared to the case of 4:1.

#### 3.1.3. Raman Spectroscopy

The Raman spectra of the a-BCN film are shown in Figure 7. In each sample, a broad peak with an overlapping graphite G-band (1580 cm^−1^) and D-band (1350 cm^−1^) is observed. In the sample with a boron and nitrogen ratio of 4:1, the G peak position tends to shift to the lower wavenumber side, and the intensity of the D peak tends to increase with a decrease in the carbon content. In addition, when the carbon content is <80%, the photoluminescence (PL) component increases, and the G and D peaks are buried in the PL background. According to previous studies, the PL component in the Raman spectrum of an amorphous carbon film is attributed to structural defects [34]. Robertson mentioned that impurities and defects in the film lead to the formation of another energy level, and the optical excitation, in this case the PL component, was due to the interaction between these energy levels. Therefore, it can be considered that the defects in the a-BCN film increased as the carbon content decreased. On the other hand, for samples with a boron to nitrogen ratio of 2:1 or 1:1, a peak peculiar to amorphous carbon is observed even when the carbon content is <60%. Thus, it can be concluded that it is necessary to consider the boron to nitrogen ratio in order to synthesize and control its structure.

Gaussian fitting was performed on the obtained Raman spectrum, and the G peak position, FWHM(G), and the I(D)/I(G) ratio were derived. Figure 8a shows an example of the fitted spectrum. As shown in Figure 8b, the sample with a carbon content ≤5% could not be fitted. Therefore, peak fitting was performed only for samples with carbon content ≥ approximately 30%. Figure 8c shows the peak fitting results with respect to the carbon content. Although these data were obtained from various a-BCN films with different elemental ratios and structures, they are all plotted on almost the same line. The G peak position gradually decreases as the carbon content increases, but it increases sharply when the carbon content is higher than approximately 80% and approaches that of the a-C film. As for the FWHM(G) and I(D)/I(G), they maintain almost the same value regardless of the carbon content, but change rapidly at a carbon content of 80%, and then approach that of the a-C film. From these results, it can be considered that the structure of the a-BCN film changes continuously with respect to the carbon content and is not affected by the boron/nitrogen ratio. From the results in Figure 4, it was thought that the range of carbon content would differ depending on the ratio of boron and nitrogen. These Raman spectroscopic analysis results also strengthen this conclusion, i.e., the ratio of boron and nitride limits the amount of carbon that can be incorporated into the film. If there is a defect in the film structure, the G peak position and the I(D)/I(G) value increase, whereas FWHM(G) decreases [35,36]. This has also clarified that this Raman parameter shows an opposite tendency when the film quality is improved by the ion assist method [37] or by increasing the incident energy of ions [38], i.e., the G peak position and the I(D)/I(G) value decrease, whereas FWHM(G) increases. Therefore, the structure of a-BCN deteriorates at low carbon contents, which affects the quality of the film.

### 3.2. Mechanical Properties

#### 3.2.1. Indentation Hardness

Figure 9a shows the indentation hardness of the a-BCN films with respect to the carbon content. The a-C film exhibits the highest hardness (23.6 GPa), and the hardness decreases linearly with the decrease in the carbon content. The hardness is almost constant when the carbon content is <90%. Figure 9b shows the Raman parameters in correlation with the hardness. In the region where the carbon content is 90% or more, the Raman parameters show almost the same value regardless of the hardness, indicating that these samples have similar structural characteristics. In other words, the mechanical properties of a-BCN film are also significantly affected by the carbon content and its structure. Recently, the correlation between carbon content in the a-BCN film and its properties have been widely discussed. For example, Qin et al. reported that the PL spectra of the BCN nanosheets are dependent on their chemical composition, and it is therefore probable that the excessive carbon atoms form a graphite phase and the nanosheets with the composition of B_0.38_C_0.27_N_0.35_ show intense emission at 3.27 eV with very weak defect-related emission [39]. Nascimento et al. points out the possibility that by doping boron or nitrogen to graphene the doping-induced band gap can vary over an order of magnitude depending on the placement of the boron and nitrogen atoms [40]. Wang et al. and Wu et al. reported that boron-doped graphene exhibits a p-type doping behavior with a considerably high carrier mobility [41,42]. Rani et al. reported that isomers formed by choosing different doping sites differ significantly in relative stability and the band gap introduced, and these aspects depend, to a great extent, upon the position of the boron and nitrogen atoms in the heterostructure [43]. However, these studies are mainly focused on the electrical or nanomechanical properties of a-BCN film. On the other hand, the results shown in Figure 9 indicate that mechanical properties can also be also discussed from the correlation with carbon content. As indicated by the FTIR results (Figure 5), the BCN films were mainly composed of B–N, C–B, and N–C bonds. The h-BN bond has been reported to be a “soft” phase with a graphite-like structure [44,45,46]. Therefore, a higher content of h-BN bonds will adversely affect the mechanical properties of the films. In contrast, C–B and N–C bonds are “hard” bonding structures, which contribute to the enhancement of hardness. As reported by Byon et al. [47] and Chien et al. [48], the C–B and N–C bonds play an important role in determining the mechanical properties by enhancing the global connectivity of the network. Furthermore, it is also well known that the hardness of the film is linearly related to the residual stress in the film [49,50]. The addition of boron and nitrogen to the a-C film enhances the hBN bond and decreases the C–B and N–C bonds or internal stress. In fact, Franceschini et al. observed that the internal stress considerably decreases with the increase in nitrogen content owing to the change in the coordination number [51].

#### 3.2.2. Frictional Properties

Figure 10a shows the evolution of the friction coefficient of the a-BCN films as estimated from the ball-on-disk wear tests. The a-C film showed the lowest friction coefficient and stable behavior, and the friction coefficient increases as the carbon content decreases. In particular, a sharp increase in the coefficient of friction is observed when the carbon content is <90%. From this result, it is considered that the sliding characteristics are strongly influenced by the structural characteristics of the film and the indentation hardness. In fact, the sample with a carbon content of 42.5% showed higher friction than the silicon substrate. Furthermore, the friction coefficient of the sample with a carbon content of 87% is almost the same as that of the silicon substrate. Thus, it can be concluded that the sliding properties of the silicon substrate can be improved by coating a-BCN film with a carbon content ≥90%. Figure 10b shows the evolution of the friction coefficient of samples with a carbon content >90%. All the samples show stable behavior with a low friction coefficient of approximately 0.1 to 0.2, which is close to that of the a-C film. Figure 11 shows the specific wear rate of each sample. The a-BCN films with a carbon content >90 at.% show similar or lower values of wear rate than the a-C film. However, when the ratio of boron and nitrogen became 1:1, the wear resistance of the film deteriorated. This is because when the ratio of boron and nitrogen becomes 1:1, the amount of B–N bond in the film structure increases, and the film hardness becomes small. Figure 12 shows the optical microscope images of the wear track and the ball after sliding. A black transfer film is observed in all the samples. As discussed in many previous studies, the low coefficient of friction and wear volume of the a-BCN film can be attributed to this “black transfer film”. According to Godet and coworkers [52,53,54], friction and wear behaviors can be discussed using third-body processes. Third bodies, in this case “black transfer films”, are formed by the relative motion of the two parent materials in the sliding contact. They often appear first as thin films transferred to the stationary counterface and then evolve into wear particles [54,55,56,57]. By separating the two parent materials, third bodies play a role in distributing the stresses and accommodating the relative motion between the counterfaces.

Raman spectroscopic studies were performed to analyze the structure of the transfer film. The red spot in Figure 12 is the focal position of the laser used for Raman spectroscopy. As shown in Figure 13, the position of the G peak shifts to a higher wave number, the FWHM(G) decreases, and the I(D)/I(G) increases after sliding for all the counterface surfaces compared to the as-deposited films. This tendency is particularly noticeable in the film on the ball surface; thus, it is considered that the black transfer film is graphitized. In addition, the tendency of graphitization was greater in the a-BCN film than in the a-C film. It has been reported that a-BCN films containing trace amounts of boron show better tribological properties for iron-based materials than a-C films under wet conditions because the graphitization of the transfer film progresses on the surface of iron-based material [58]. In general, the sliding properties of a-BCN films against iron-based materials depend on the relative humidity. Under a low humidity (5% RH) environment, a-BCN films show a relatively high coefficient of friction (approximately 0.4 to 0.8) owing to the wear caused by boron carbide. On the other hand, under a relatively mild humidity (45% RH) environment, they shows a low coefficient of friction (approximately 0.1) because of the formation of a boric acid layer by the reaction of boron in the a-BCN film with water vapor in the atmosphere [59]. In this study, because the ball-on-disk test was conducted under a mild humidity environment (approximately 25% RH), there was a possibility of formation of the boric acid layer, as shown schematically in Figure 14.

Finally, XPS studies were carried out to confirm the formation of boric acid. Figure 15 shows the X-ray photoelectron spectra of the a-C and a-BCN films. Two types of relatively large peaks of C 1s and O 1s and small peaks of N 1s and B 1s are detected for the a-BCN film. The O 1s peak of a-BCN is sharper than that of a-C, indicating that the former contains more oxygen. The origin of the O 1s signal for the a-C sample is attributed to the contamination of the film after exposure to the atmosphere. Figure 16 shows the results of peak fitting of the B 1s, C 1s, N 1s, and O 1s spectra of the a-BCN film. Each peak was deconvoluted separately [20,22,29,30,31,32,33,60,61,62]. The O 1s and B 1s spectra of a-BCN confirm the presence of B–O bonds. This suggests the possibility of the presence of a boric acid layer at the sliding interface.

## 4. Conclusions

a-BCN thin films were successfully prepared with carbon contents in the range of 40–100% using a newly developed film coating setup, which combines vacuum arc vapor deposition and magnetron sputtering.

The boron/nitrogen ratios of the films were 4:1, 2:1, and 1:1. The hardness and the friction coefficient of the a-BCN thin film deteriorated as the carbon content decreased, regardless of the boron/nitrogen ratio.According to Raman spectroscopy studies, the changes in the mechanical properties are attributed to the deterioration of the film structure when the carbon content is < approximately 90 at%. This was confirmed by the increase in the G peak position and the I(D)/I(G) ratio and the decrease in the FWHM(G) at carbon contents <90 at%.By adding a small amount of boron and nitrogen, while ensuring that the carbon content does not fall below 90%, it is possible to prepare a-BCN film with a hardness almost identical to that of the a-C film and a wear resistance superior to that of the a-C film.

The results of the present study can provide new insights for the future development of super-wear-resistant coatings. In the future, to synthesize a-BCN film with a much higher hardness and wear resistance than those of the samples in this study, the energy of the ions incident on the substrate must be significantly increased to make the structure of c-BN and BC_2_N phase. For that purpose, the development of a bias-voltage application technique synchronized with the discharge cycle of the arc plasma gun is indispensable.

## Figures and Tables

**Figure 1 materials-14-00719-f001:**
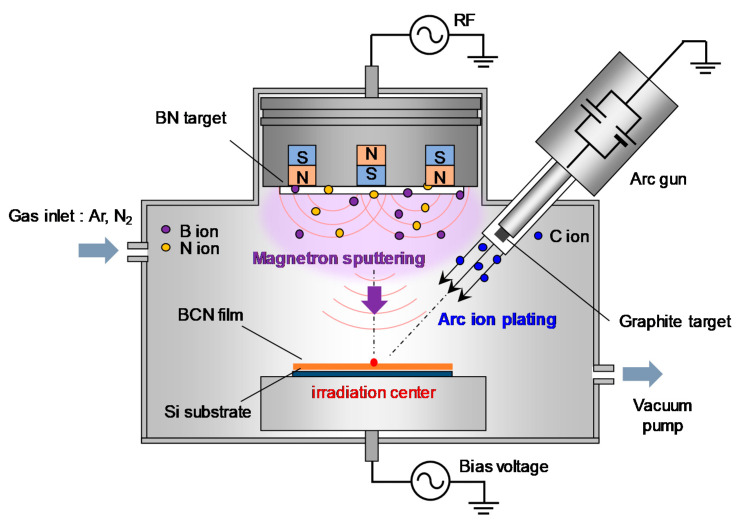
Schematic of the arc-sputtering hybrid coating setup.

**Figure 2 materials-14-00719-f002:**
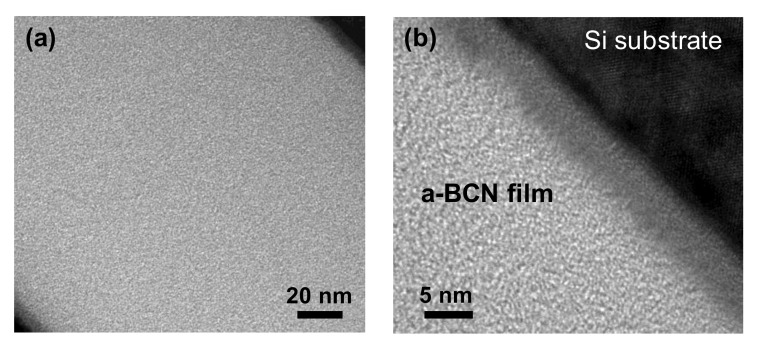
Cross-sectional TEM image of the a-BCN film: (**a**) Over view of the a-BCN cross section (low magnification); (**b**) Close up view of the a-BCN cross section (high magnification).

**Figure 3 materials-14-00719-f003:**
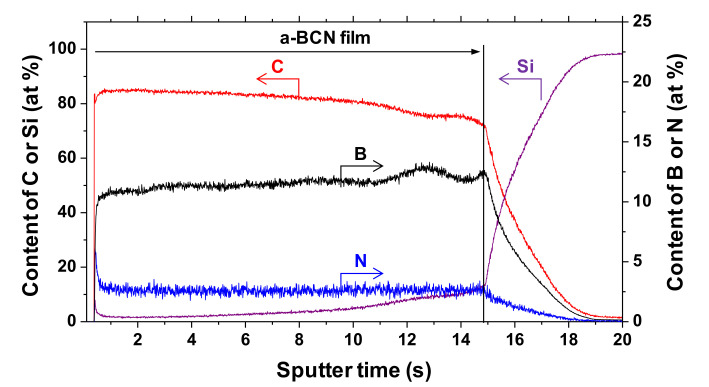
Glow discharge optical emission spectroscopy (GDOES) depth profile of chemical composition of the a-BCN film.

**Figure 4 materials-14-00719-f004:**
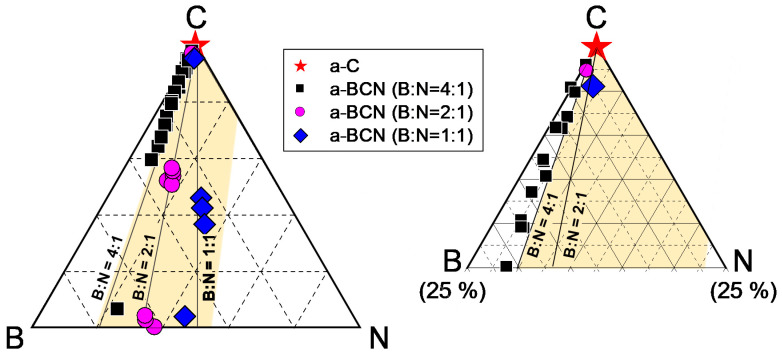
B–C–N ternary diagram showing the elemental composition of the a-BCN film.

**Figure 5 materials-14-00719-f005:**
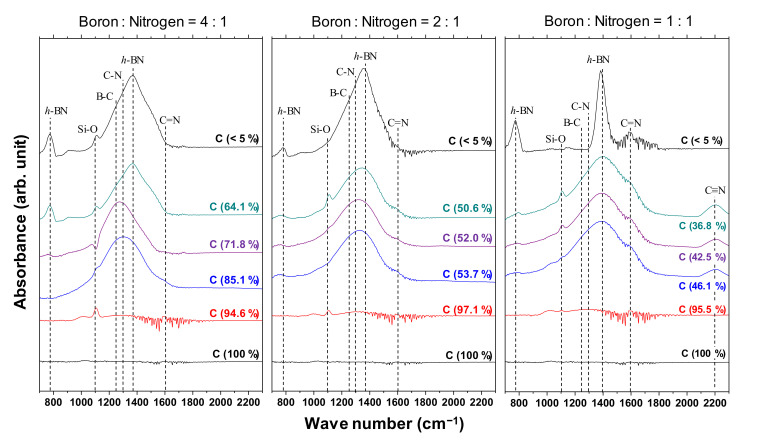
IR spectra of the a-BCN films.

**Figure 6 materials-14-00719-f006:**
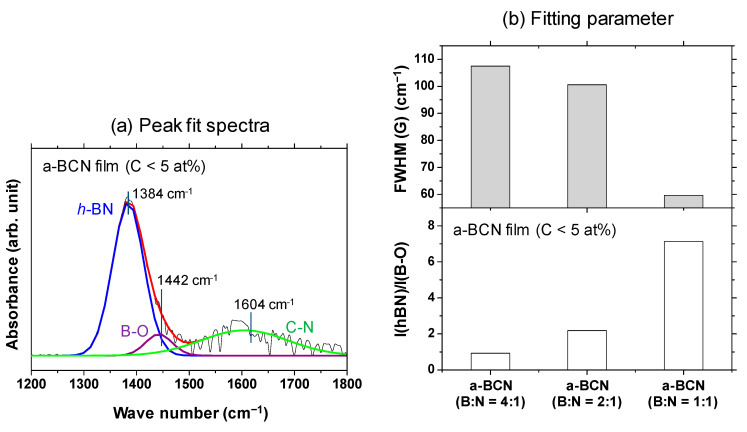
(**a**) Deconvoluted FTIR spectrum of the a-BCN film (C < 5 at%) with boron and nitrogen ratio of 1:1. (**b**) I(hBN)/I(B–O) and FWHM (hBN) of the a-BCN films (C < 5 at%).

**Figure 7 materials-14-00719-f007:**
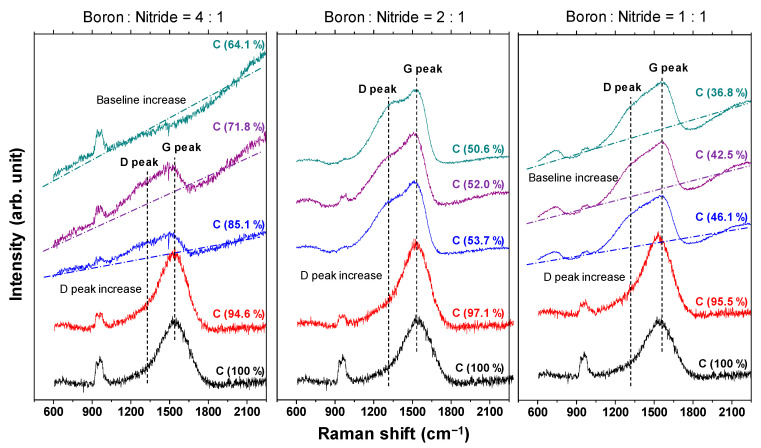
Raman spectra of the a-BCN films.

**Figure 8 materials-14-00719-f008:**
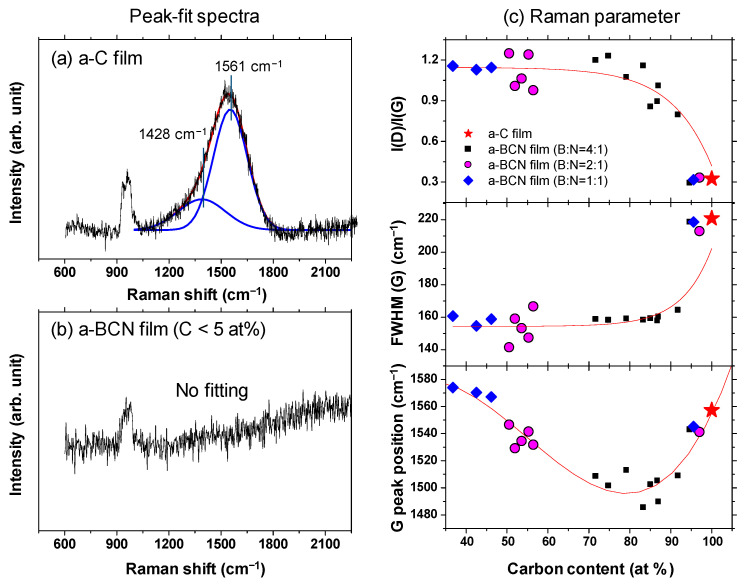
(**a**,**b**) Deconvoluted Raman spectra of the a-BCN films. (**c**) Raman parameters of the a-BCN films with respect to the carbon content.

**Figure 9 materials-14-00719-f009:**
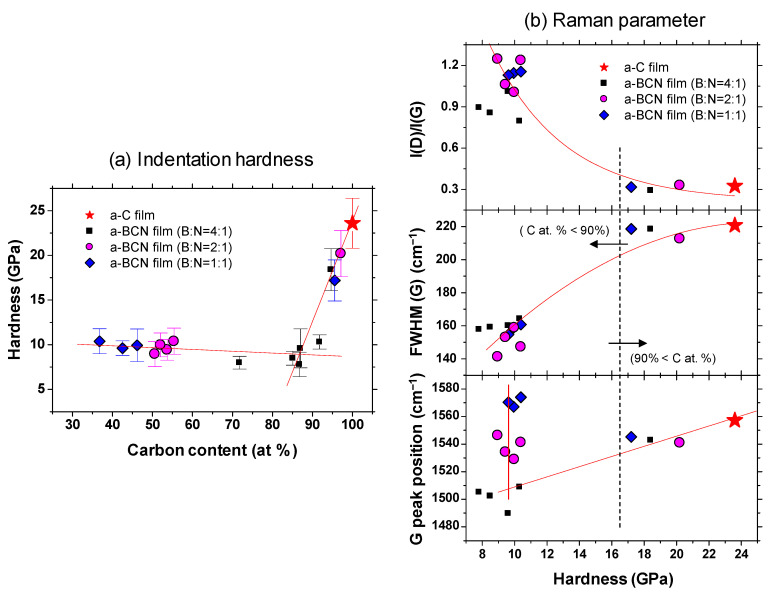
(**a**) Indentation hardness as a function of the carbon content; (**b**) Raman parameters with respect to the indentation hardness.

**Figure 10 materials-14-00719-f010:**
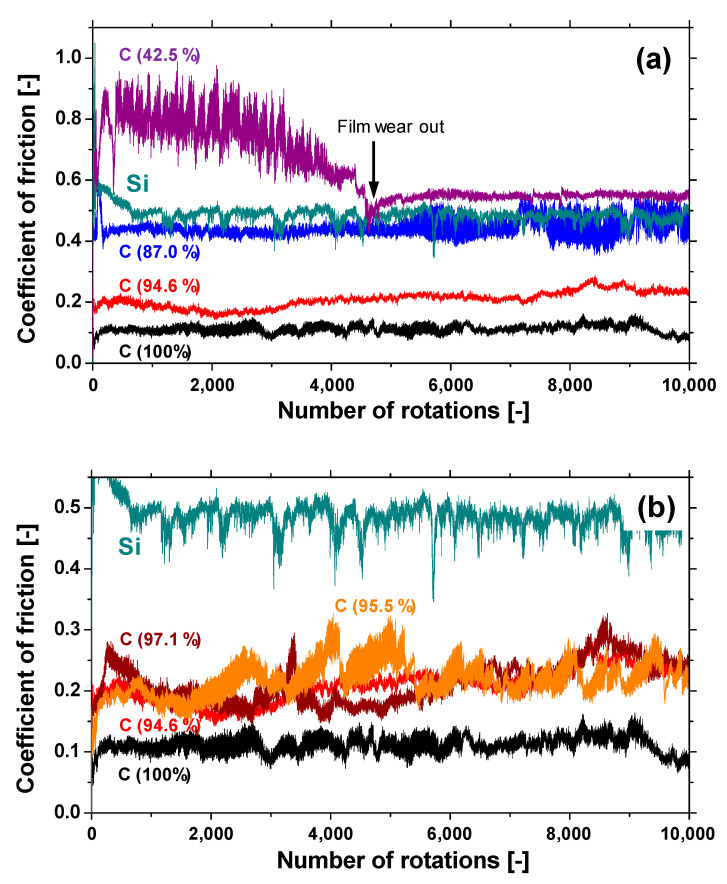
(**a**) The evolution of the friction coefficient of samples with different carbon content. (**b**) The evolution of the friction coefficient of samples with a carbon content >90%.

**Figure 11 materials-14-00719-f011:**
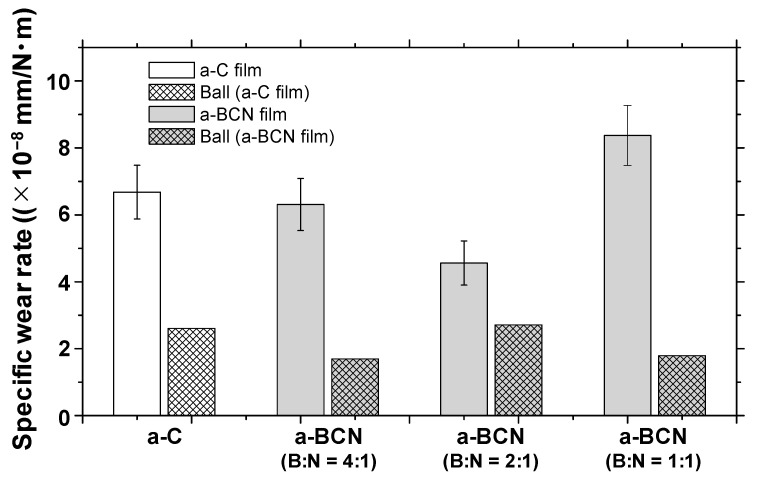
Specific wear rates of a-BCN and a-C films.

**Figure 12 materials-14-00719-f012:**
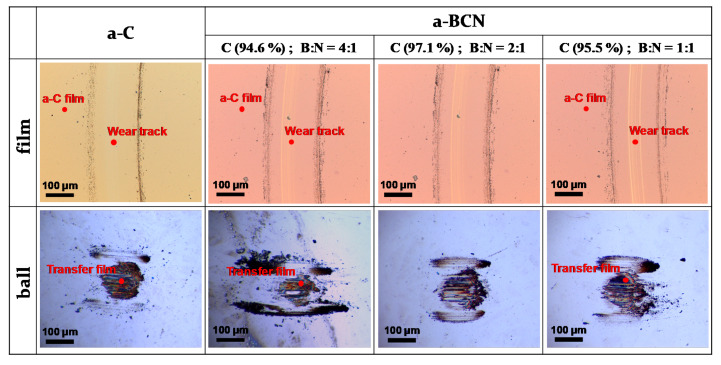
Optical microscope images of the wear track on the film and the wear spot on the ball sample.

**Figure 13 materials-14-00719-f013:**
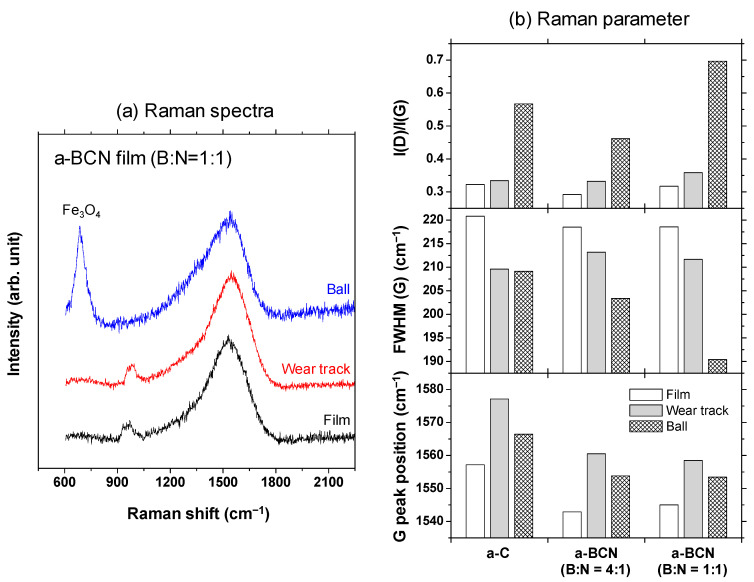
(**a**) Raman spectra of the surface of the as-deposited film, wear track, and ball sample and the corresponding (**b**) Raman parameters.

**Figure 14 materials-14-00719-f014:**
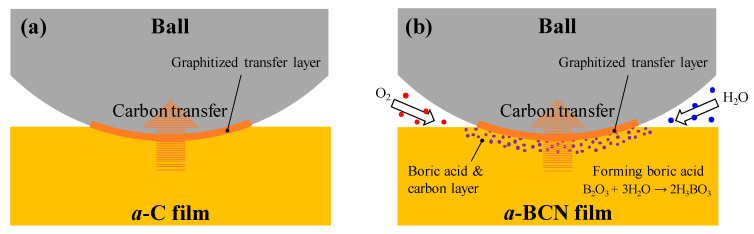
Schematic diagram of the wear behavior during sliding of (**a**) a-C film and (**b**) a-BCN film.

**Figure 15 materials-14-00719-f015:**
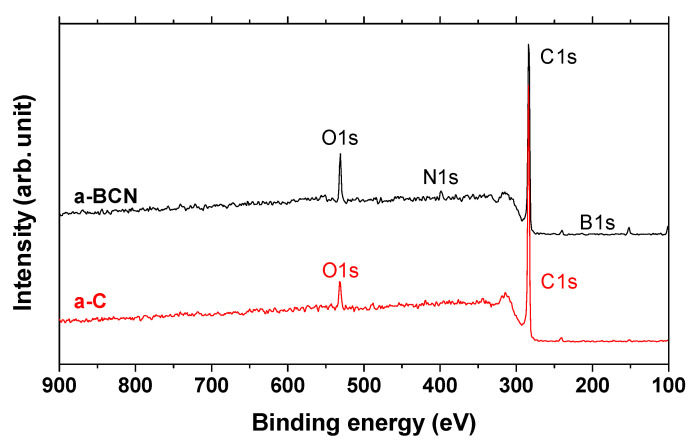
XPS survey spectra of a-C and a-BCN films.

**Figure 16 materials-14-00719-f016:**
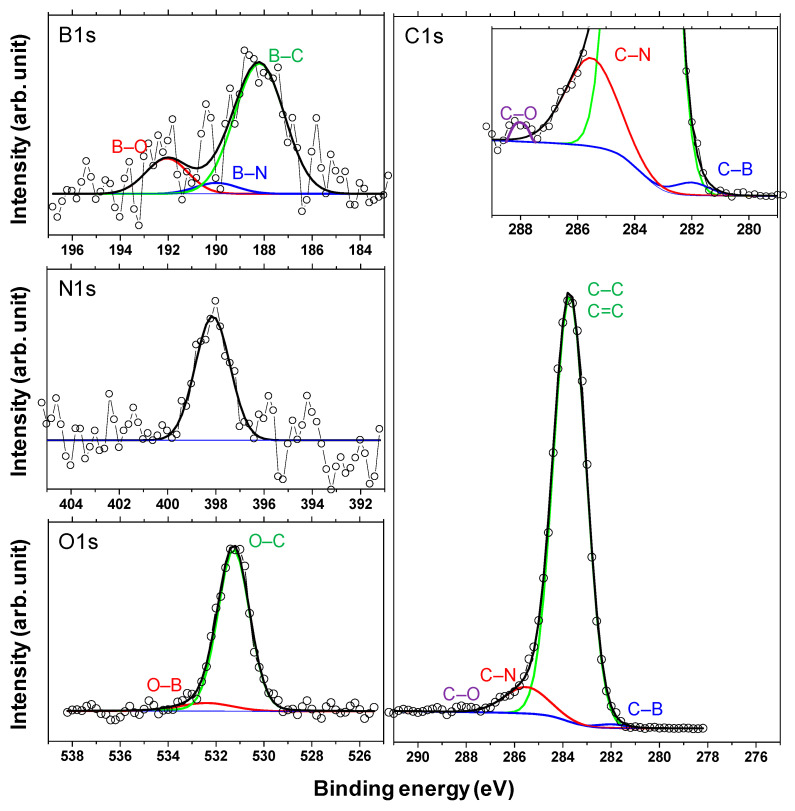
Deconvoluted B 1s, N 1s, C 1s, and O 1s X-ray photoelectron spectra of the a-BCN films.

**Table 1 materials-14-00719-t001:** Deposition conditions for amorphous carbon (a-C) film and amorphous boron carbon nitride (a-BCN) films.

		a-C	a-BCN
Pressure		0.4 Pa	
Ar; N_2_ flow		10 cm^3^/min; 0 cm^3^/min	10 cm^3^/min; 0 cm^3^/min 10 cm^3^/min; 4.3 cm^3^/min 10 cm^3^/min; 10 cm^3^/min 0 cm^3^/min; 10 cm^3^/min
Arc gun	Target	Graphite	Graphite Boron (0.5%) doped Graphite Boron (1.0%) doped Graphite Boron (5.0%) doped Graphite Boron (20%) doped Graphite
	Capacitance	360 µF	
	Arc voltage	−100, −400 V	
Magnetron sputtering	Target	―	hBN
	RF power	―	100, 200, 300 W
Substrate potential		floating, DC −100 V	
Deposition time		60, 180 min	

## Data Availability

No new data were created or analyzed in this study. Data sharing is not applicable to this article.

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
