# Peer review of "Structural and Mechanical Properties of a-BCN Films Prepared by an Arc-Sputtering Hybrid Process"

_materials, 2021, doi:10.3390/ma14040719_

Round 1

Reviewer 1 Report

This paper examines a-BCN films prepared for arc sputtering using different deposition conditions. The results obtained are very interesting and fit the purpose of this research. The authors explained in an exhaustive way the deposition technique used and the properties of the obtained films. Furthermore, the various measurement techniques used to characterize the deposited films were explained. The authors indicated that the carbon content influences the properties of the deposited films and that by controlling the Boron and Nitrogen content during the deposition of the films it is possible to synthesize films with good properties very similar to those found in amorphous carbon, but with improved wear. This result is very interesting and paves the way for the development of new super wear resistant coatings.

Reviewer 2 Report

In this article by Hirata et al., the authors studied the structure and mechanical properties of synthesized a-BCN films. The results are interesting and fit the scope of Materials. However, some issues must be addressed before the paper can be recommended for publication. Please see the suggestions below:
1) The newly developed deposition process of the films is very interesting. Could the authors clarify if the deposition stage is rotating? Provided that it is stationary, it appears that if carbon is deposited from the side, one can expect the inhomogeneous composition of the film. Please comment on this issue.
2) Formatting of Table 1 is confusing. Please decide whether to align certain entries to the center or to the side and do it consistently.
3) TEM acceleration voltage nor reported.
4) The description of FTIR experiments are unclear whether the samples were dried to remove the possible influence of moisture.
5) Were the Raman spectra obtained only in one location per sample?
6) There are two panels (b) in Fig. 2.
7) Please verify if it should not read "Boron: Nitrogen" above Fig. 5.
8) Is it possible to do the same experiment for several locations (Fig. 6a), which would produce error bars for Fig. 6b?
9) Unexpected empty space throughout text should be removed e.g. Pages 7-9.
10) Again no error analysis in Figs. 8b, 9, 11, 13. It is important to study the uncertainties to ensure that the interpretation of the results is justified.
11) Headlines should not be separated from the corresponding sections (Line 257).
12) Please provide a future outlook section in the Conclusions section.

13) "According to Hirata et al., if there is a defect in the film structure, the G peak position and the I(D)/I(G) value increase, whereas FWHM(G) decreases 
[35–38]. " (Lines 245-247). Please kindly consider if citing four papers of the first author to support a single claim is absolutely necessary. 

Reviewer 3 Report

The article is interesting. Authors should carefully study the comments and make improvements to the article step by step. After major changes can an article be considered for publication in the "Materials".

The current paper investigates a-BCN films manufactured using arc sputtering using different coating conditions. The authors aim is to explain the relationship between the film structure and characteristics and the measured mechanical properties. The authors use some inspection devices to check for surface and microstructure properties. Destructive testing such as nano indentation was used to measure the hardness of the coating and friction tests were used to measure the friction and wear. The authors conclude that the carbon content can highly influence the film properties, the authors also claim that by controlling other elements in the film such as N and B can obtain good film properties  similar to those found in amorphous carbon and with enhance wear properties.

Line 48 and 45 the authors must avoid using bulk citations without giving them enough credit. Either reduce them or discuss each one of them in details, please check this issue all over the manuscript

Literature review is non existent, it is very generic and does not explain the problem in hand. The authors must carry out a proper literature review indicating previous studies on similar subject and discuss what has been done and what was found, then highlight what their work bring differently to past research.

Line 67 remove the word so far, the authors are encouraged to use scientific writing style all over the manuscript, please check everywhere else for similar issues.

For table 1, regarding the parameters used in the study for the film deposition, why the authors choose those specific values, were they chosen based on the machine range or are they recommended values for fabricating such films in industry, either way please explain and discuss.

Line 108 why specifically 2 minutes? Or do you mean was etched until all surface contamination was removed? Perhaps it is better to rephrase such sentences to remove ambiguity about the accurate timing for such processes.

Line 159 can you support this claim by a reference?

The authors must avoid using we, our or similar in all the manuscript. Please check carefully and amend.

Did you measure the surface roughness of the films?

Section 3.1.1 there is no discussion of results bur rather describing what is seen in the graphs.

Line 185-190 please avoid bulk citations. Also the references added in this section are not clear why they are there, are you explaining a phenomena or a results, if yes then refer to what those papers found and explain in details and compare with your current findings.

Line 197 why this phenomenon happens? Please explain in more details and support with references if possible. Also when you say peaks are sharper, how much sharper they are compared to the other ones observed?

Line 227 “that the defects in the a-BCN film increased as the carbon content decreased” please explain this and support with references.

Line 284, in which way it affects the film quality, please explain further or give examples.

Line 267 too many references, please give credit to others work instead of bulk citations style.

Section 3.2.2 is well discussed, please address previous sections in similar way.

In figure 11, the trend appears to be nonlinear, why for example the wear rate decrease then increase, please explain and support with references.

For conclusion, it is recommended to use bullet points, one for each section discussed previously in order to make it easier for the readers to follow what has been done in the paper however this is up to the authors.

Round 2

Reviewer 2 Report

Thank you for providing convincing arguments. I recommend the paper for publication.

Reviewer 3 Report

The authors have answered all questions, the paper can now be accepted.